# Research on Speed Planning in the Constant Speed Interval and Quality Prediction on the Transition Surface

**Jichun Wu [1,*], Ke Xu [1], Huicheng Zhou [2] and Dapeng Fan [3]**

[1] School of Mechanical Engineering, Xiangtan University, Xiangtan 411105, China; 202021002123@smail.xtu.edu.cn
[2] National NC System Engineering Research Center, Huazhong University of Science and Technology, Wuhan 430074, China; zhouhuicheng@hust.edu.cn
[3] School of Intelligent Science, National University of Defense Technology, Changsha 410015, China; fdp@nudt.edu.cn
[*] Correspondence: wujichun@xtu.edu.cn

**Abstract:** Aiming at the particularity on the transition surface, this paper carried out research on adaptive speed planning, local optimization of machining toolpaths to improve the surface quality of the transition surface, and visualization of motion parameters to predict the surface quality of the transition surface. Constant speed interval planning and trajectory optimization were carried out on the transition surface between the bottom of a Cola bottle and the curved surface with variable curvature, respectively, and the visualization maps before and after optimization were obtained. The comparison showed that the motion parameters of the optimized visualization map had better smoothness. Further, the effectiveness of the estimated surface quality was verified through the actual machining of the variable curvature transition surface and Wuzhi mountain parts. The consistency of the improvement of the processed surface quality and the quality of the visualization were verified.

**Keywords:** quality prediction; Akima fitting; the constant speed interval; visualization





## 1. Introduction

Improving the efficiency and quality of high-speed machining are eternal themes in numerical control machining, and researchers are constantly exploring how to balance the relationship between them, such as the surface quality and efficiency of parts in mold machining. Moreover, the surface often contains the transition surface [1]. Although the transition surface is a very general problem in theory, it has important significance and practical value in many fields of practice. In order to construct a variety of products in product modeling and manufacturing, and make it possible to conduct NC machining, it is necessary to connect and transit between the transition surface of variable curvature and the conventional surface or plane. Therefore, transition surface modeling and machining technology have been widely studied and applied.

Transition surface is one of the important branches of CAD/CAM, which is a smooth surface between different surfaces under certain conditions. At present, in the modeling stage, with the help of UG, ProE and other three-dimensional software, smooth and continuous modeling design can be handled well. However, in the manufacturing stage, because of the particularity of transition surface, it is often more difficult to control the transition surface than the ordinary surface or plane connected with it.

Under the same machining conditions, that is, the machine tool is set with the same machining parameters, and the same tool and machine tool have the same dynamic constraints. This paper focused on the special application object of the transition surface. Due to the fact the tool path forms different geometry as a result of curvature change, large machining damage can be caused by machining of the transition surface. If precise speed planning is carried out for each point of the trajectory, it may produce a large speed fluctuation, and

the smaller difference in the trajectory will be amplified into a larger difference in speed. In the process of motion control, the speed difference will be further enlarged to the difference of following error, which will lead to the discontinuity of the final surface morphology, forming visible defects. In order to improve the machining quality of transition surface, we conducted some research on aspects such as speed planning, local optimization of machining trajectory and motion parameters visualization to predict the surface quality of the transition surface.

Conventional acceleration and deceleration methods are usually used for speed planning of transition surface. Speed planning originated from classic S-type acceleration and deceleration in [2–5]. According to the machining parameters, such as the distance of machining trajectory and the speed of starting point and end point, different acceleration and deceleration modes are planned [6–8]. According to the special points of the machining trajectory, including inflection points, extreme points and other geometric information, the dynamics of the machine tool, and the characteristics of the tool, the precise speed planning of the machining trajectory is carried out [9–16].

Aiming at the machining trajectory after speed planning, trajectory optimization can be carried out in different stages. Usually, geometric optimization and an interference test are carried out in the CAM stage. After post-machining, the G code of CNC operation is generated and the main expression trajectory of the G code is small line segment or small line segment mixed with circular arc. Therefore, trajectory optimization can generate a spline trajectory through offline fitting after the small line segment G code, and is inputted into the CNC system for machining [17–19]. Spline trajectories are more and more widely used in NC systems because of their inherent advantages compared with small line segment trajectories. Small line segment trajectories are usually fitted into spline trajectories globally. For the free surface with transition surface, the transition surface is the most difficult to control. Therefore, the local optimization of the tool path is adopted, and the G code from CAM is used to directly machine the other surfaces, while the spline trajectory is generated by the local optimization of the transition surface. Finally, the adaptive speed planning and local optimization of the tool path are used to control the trajectory of the transition surface [20–22]. As a spline tool path, NURBS is widely used for its advantages of high continuity, convenient local adjustment and good shape retention during fitting [23–26]. However, in NC interpolation, the analytical expression is an iterative difference expression, resulting in a large amount of real-time calculation. The Akima curve is a piecewise cubic polynomial curve with continuous first-order derivative established between known data points, and its generation process is simple and easy to calculate. The generated curve is smooth and natural, and the segments can guarantee continuity above G1 [27–34].

The discontinuity of geometric properties and motion instructions will lead to pits and bulges of the transition surface topography. The continuity of the actual speed, acceleration and other motion parameters generated dynamically can only be analyzed by the real-time data recorded in the machining process. Quality estimation of the transition surface before the actual machining can reduce the work of subsequent surface quality testing. Usually, through the method of trial machining to test the surface quality of parts, if the quality does not meet the requirements, which would lead to waste of motion parameters in the machining process, the machining parameters are modified or the machining code is optimized. However, this method of checking the surface quality is fussy and time-consuming. In fact, when the code is loaded into the NC system to simulate the machining, the running machining parameters contain state information. However, the existing machining methods do not save these machining parameters and waste the important information needed for machining quality evaluation. Even if the machining parameters are saved, how to establish the mapping with the machining CL points and evaluate the machining quality are key problems to be solved. Aiming at the particularity that the transition surface is often more difficult to control than the ordinary surface, adaptive speed planning, local optimization of machining toolpaths to improve the surface quality of the transition surface, and visualization of motion parameters to predict the surface quality of the transition

surface are studied in this paper. There is no need for repeated trial machining, and the parameters can be visualized to accurately predict the surface quality so as to improve the machining quality and efficiency of the transition surface by using the information contained in the motion parameters and process parameters of the NC device.

## 2. Speed Planning in Constant Speed Interval

Since the tool path of the transition surface is a variable curvature interval, traditional speed planning at each point will produce large speed fluctuations. Based on this, the concept of "constant speed interval" is proposed, and the average curvature is calculated in the tool path interval of the transition surface, and the planning speed of the interval is further calculated with the average curvature, so as to realize the motion planning of the tool path of the transition surface. In the G code, a section of tool path is obtained according to the characteristics of the direction-parallel cutting machining and contour-parallel cutting machining. In the contour-parallel cutting toolpath, the blue CL point is the boundary point of adjacent trajectories, and a section of tool path is obtained, as shown in Figure 1a. When it is the direction-parallel cutting tool path, the boundary points of adjacent trajectories are used to distinguish the points with similar characteristics and two adjacent (nearest interval) tool paths, and a section of tool path is obtained through boundary points, as shown in Figure 1b. For the toolpath without boundary, as shown in Figure 1c, the distance threshold can be set through the cumulative sum of small line segment distances. When the cumulative distance is greater than the distance threshold, the previous CL point is the boundary point, and the tool path is segmented. Therefore, no matter what kind of tool path it is, it is first segmented, and each segment includes the CL points of the ordinary surface, and may also include the transition surface. Further, in each tool path, the CL point of the transition surface, especially the starting point and end point, have speed planning that is suitable for them planned in the interval of the transition surface. In order to accurately identify the CL points in the interval of the transition surface, topological geometric modeling is carried out for each CL point firstly, and then the curvature of the CL point, the distance ratio and the curvature ratio between the -***characteristic CL point and the adjacent CL points are calculated.

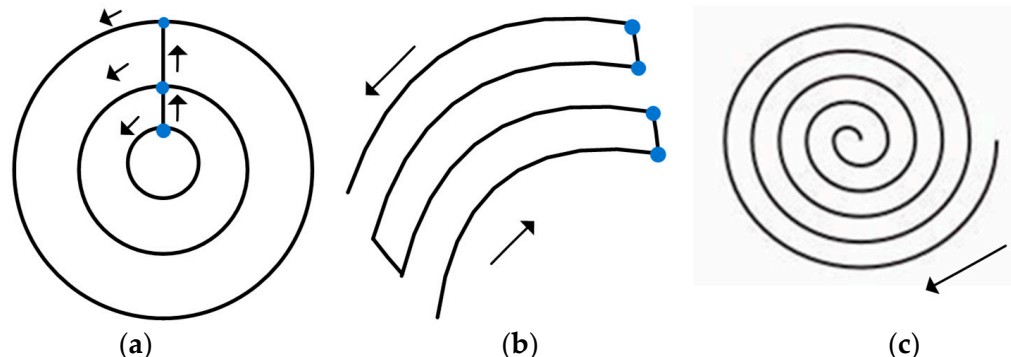

**Figure 1.** Toolpath classification (**a**) contour-parallel toolpath (**b**) direction-parallel toolpath (**c**) no boundary toolpath.

After calculating a section of tool path, it is necessary to undertake speed planning for the tool path. In traditional speed planning for the tool path, every CL point for precise planning will produce a larger speed fluctuation, according to the characteristics of the transition surface with variable curvature. Although the curvature changes, the variation range is small and it is a gradual process. Therefore, it is very necessary to develop adaptive speed planning in the constant speed interval.

The curvature of each original data point is solved by forming an approximate circular arc with three adjacent data points [35]. As shown in Figure 2, if the number of original data

points is $n+1$, the curvature radius corresponding to the ith data point is $\rho_i (i = 0, 1, \cdots, n)$, then the radius of curvature can be approximately obtained by Equation (1).

$$\rho_i = \frac{|Q_{i+1} - Q_i| \cdot |Q_{i-1} - Q_i| \cdot |Q_{i+1} - Q_{i-1}|}{2|(Q_{i+1} - Q_i) \times (Q_i - Q_{i-1})|} \tag{1}$$

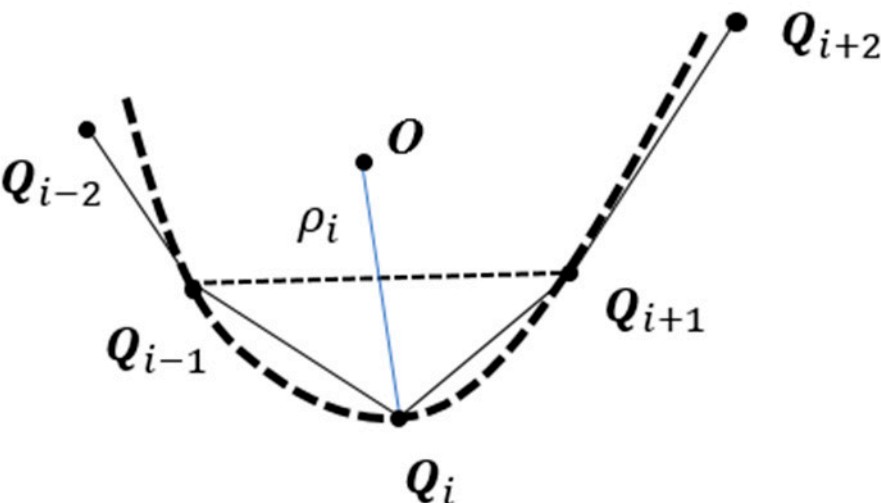

**Figure 2.** Curvature radius calculation.

As shown in Figure 3 below, the error model of characteristic CL points identification is first established according to the fitting error to obtain the contour shape of the trajectory expressed by the minimum CL points, and these CL points include the starting point and the end point of the constant speed interval. Further, in order to obtain a more accurate constant speed interval. The curvature of each CL point is calculated, the threshold values of distance ratio and curvature ratio are set, and the distance and curvature ratio between characteristic CL point and the adjacent CL points are calculated. When the threshold is exceeded, it is the starting point or end point of the constant speed interval, otherwise it is the middle point of the same speed interval, and incorporated into the constant speed interval. The remaining characteristic CL points are the starting point or the end point of the constant speed interval. The specific algorithm flow is as follows:

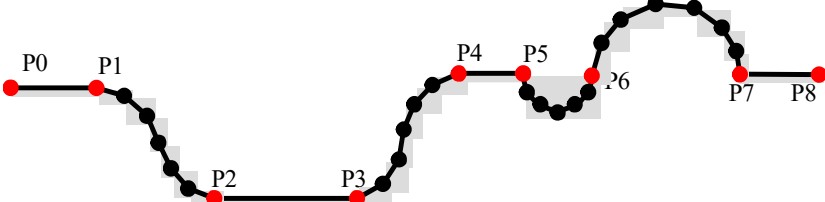

**Figure 3.** Constant speed interval planning of tool path.

Set error threshold $E_{thr}$, Distance ratio threshold $L_{thr}$, Curvature ratio threshold $Q_{thr}$;

(1) In G code, a section of tool path is obtained according to the boundary point;
(2) According to the error threshold $E_{thr}$, calculate the characteristic CL points;
(3) Calculate the curvature ratio $Q_{i-1}$ for each characteristic CL point;
(4) For each characteristic CL point, calculate the curvature ratio $Q_i$ and distance ratio $L_i$ of the front and back CL points. If $1/L_{thr} < L_i < L_{thr}$ and $1/Q_{thr} < Q_i < Q_{thr}$, Go to step (5), otherwise go to step (6);
(5) The characteristic CL point $P_i$ is the point inside the constant speed interval, which is incorporated into the constant speed interval;
(6) The characteristic CL point $P_i$ is the starting or end point of the constant speed interval;
(7) Calculate the constant speed range of the tool path and carry out speed planning.

### 3. Trajectory Optimization in the Constant Speed Interval

In NC machining, small line segments are usually used to express the tool paths, which will lead to large CL points and discontinuous trajectories. Spline trajectories are more and more widely used in NC systems because of their inherent advantages compared with small line segments. So, small line segments for tool paths are usually fitted into spline trajectories. The Akima curve is a piecewise cubic polynomial curve with continuous first-order derivative established between known data points, and its generation process is simple and easy to calculate. The generated curve is smooth, and the segments can guarantee continuity above G1. Therefore, this paper used the Akima fitting tool path algorithm, based on arc length parameters, to optimize the trajectory of the constant speed interval.

In the constant speed interval, the Akima fitting tool path algorithm was used. In this section, the tangent vector was calculated according to the starting and end CL points to ensure the convexity and concavity of the fitting curve. The CL points were parameterized with arc length information, and the tool path of Akima spline was fitted when the new CL point was input, that is, a section of tool path was generated after importing a CL point in the constant speed interval. The slope vector is calculated as follows:

$$m_{ix} = \frac{x_{i+1} - x_i}{l_i}, m_{iy} = \frac{y_{i+1} - y_i}{l_i}, m_{iz} = \frac{z_{i+1} - z_i}{l_i}, i = 1, \ldots, n \tag{2}$$

where, $l_i$ represents the arc length of adjacent CL points, so the Akima curve of arc length parameterization is:

$$r(l) = A + B \cdot (l - l_i) + C \cdot (l - l_i)^2 + D \cdot (l - l_i)^3, l \in [0, l_i] \tag{3}$$

Due to the need of calculation, there are four auxiliary slope vectors added at the starting and end CL points, and the calculation is as follows:

$$\begin{cases} m_0 = 2m_1 - m_2 \\ m_{-1} = 2m_0 - m_1 \\ m_n = 2m_{n-1} - m_{n-2} \\ m_{n+1} = 2m_n - m_{n-1} \end{cases} \tag{4}$$

The expression of the tangent vector $s(l_i)$ is calculated from $l_i$ the above equation, and is given as follows:

$$s(l_i) = \begin{cases} \frac{m_{i-1} + m_i}{2}, when\ m_{i-2} = m_{i-1} \neq m_i = m_{i+1} \\ \frac{|m_{i+1} - m_i| \cdot m_{i-1} + |m_{i-1} - m_{i-2}| \cdot m_i}{|m_{i+1} - m_i| + |m_{i-1} + m_{i-2}|}, otherwise \end{cases} \tag{5}$$

The coefficient vectors **A**, **B**, **C** and **D** of Equation (3) can be obtained simultaneously from (2), (4) and (5). The calculation expression is as follows:

$$\begin{cases} \mathbf{A} = [x_i, y_i, z_i] \\ \mathbf{B} = s(l_i) \\ \mathbf{C} = \frac{3m_i - 2s(l_i) - s(l_{i+1})}{l_{i+1} - l_i} \\ \mathbf{D} = \frac{-2m_i + s(l_i) - s(l_{i+1})}{(l_{i+1} - l_i)^2} \end{cases} \tag{6}$$

In the fitting, the calculated terminal tangent vector of the previous segment is retained as the starting tangent vector of the next segment to ensure the G1 continuity of the fitting curve. According to the arc length parameterization, the tangent vector of the end point of the next segment is calculated in this recursive way.

It is assumed that $P_1, \ldots, P_6$ are the CL points in the constant speed interval. At first, $P_1$ is taken as the starting point and $P_2$ is taken as the end point. The arc length is replaced by the chord length of $P_1 P_2$ to calculate the tangent vector of the two points and

the tool path fitted by Akima in section $P_1P_2$. The next CL point is imported, and $P_2$ is taken as the starting point. The tangent vector $s(l_3)$ of $P_3$ is calculated through arc length parameterization, and then the tool path fitted by $P_2$ and $P_3$ segment is calculated. Then, it is a case of continuing to import the CL points in the constant speed interval, and using arc length parameters to carry out recursive fitting of the Akima tool path between adjacent CL points until the end. This recursive fitting aims to realize the recursive mode by adding the CL point, parameterizing its arc length, and fitting the algorithm Akima curve of the segment. In addition, geometric information, such as tangent vector of the CL point at the end of the previous segment, is also the starting point of the next segment, so, there is no need to repeat the calculation, and improve the calculation efficiency.

## 4. Developed Visualization Software iSurface

The development of visualization software is a comprehensive improvement of the machining accuracy, machining efficiency and surface quality of parts in surface machining. The machining accuracy, efficiency and surface quality of parts are highly correlated with the control data in the machining process.

Based on the interpolation data collected in the interpolation simulation or actual machining process, this software is combined with the programming trajectory (G code) to analyze the speed and trajectory of the collected interpolation points to realize the data traceability of machining defects. The subsequent machining provides optimization direction.

The internal design of the software is divided into interface layer, command layer, data structure and core algorithm layer, including the programming trajectory analysis function module, interpolation trajectory analysis function module, global speed interval division function module, etc., as shown in Figure 4.

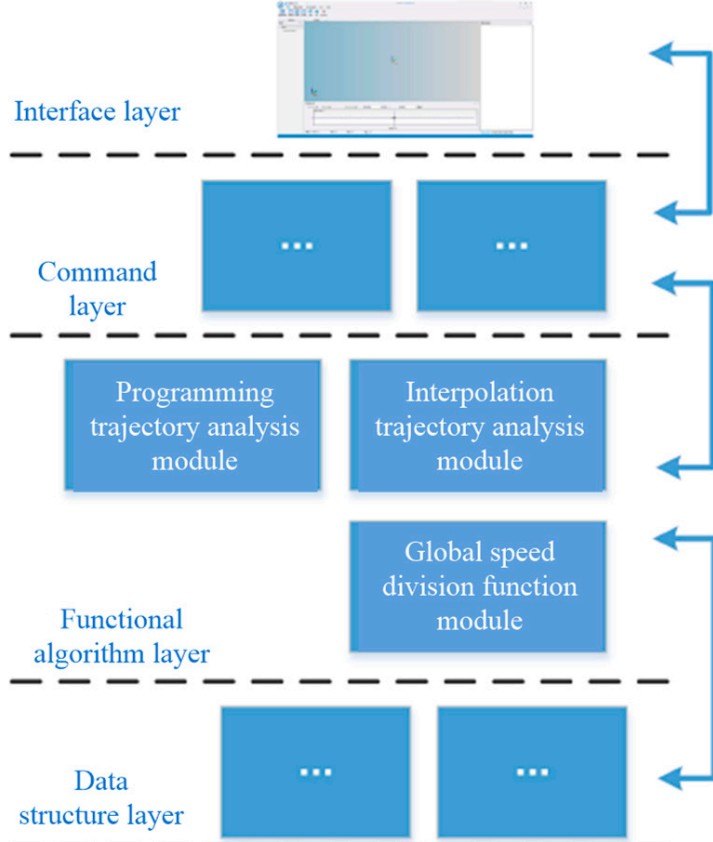

**Figure 4.** Software architecture.

Software that was independently developed to visualize the motion parameters and estimate the surface quality of the transition surface can read the binary interpolation files generated by the NC device simulation software, and can also read the original G code, which has the basic functions of visualization of motion parameters and prediction of machining surface quality. The main interface of this software consists of five parts, including the toolbar menu of the software and various operation tool buttons. There is a navigation tree for managing programming and interpolation trajectory nodes and a data bar is used to display text data of programming and interpolation trajectories. The bottom view is used to display the signal curve of the interpolation trajectory and the information of the small line segment of the programming trajectory. At present, the main map display window and the visualization map display area of motion parameters can be used to color the cutter site surface according to motion parameters, such as synthesis speed, synthesis acceleration, speed of each axis, acceleration of each axis, and dexterity of each axis. At the same time, a corresponding relationship is established between the map area of the main window and the text window on the right side, so that the line number of the corresponding machining code can be quickly found in the convex or dent area of the visualization map, and the local optimization of speed can be carried out to improve the surface quality. At the same time, by coloring the motion parameters on the surface of the CL point in the graphic display area, a visualization map of the motion parameters can be obtained, which can directly predict the machining surface quality of the parts.

- The software has the ability to view orientation of the graphical interface, including isometric, top view, front view, and left view. In the window method, a rectangular box to enlarge the area can be selected. There are shortcut keys for some of the more frequently used buttons. Hovering the mouse over the button will pop up a prompt box. The letters in brackets correspond to the keys on the keyboard.
- There are two types of imported trajectories: programmed trajectory and interpolation trajectory. The programming trajectory is imported, the programming trajectory node will be added under the navigation tree. The corresponding programming trajectory will be displayed in the graphics window, and the selected G code text file will be loaded in the programming trajectory data column. The interpolation trajectory is imported, and the actual interpolation data or theoretical interpolation data can be selected.
- The locating interpolation trajectory function can be set to display the starting point and the interpolation trajectory points. After clicking OK, the graph window, data column and signal curve will be refreshed to the interpolation data corresponding to the entered locating number. After inputting the G code line number to locate interpolation trajectory, the graphical interface will highlight the first interpolation point of the G code line number. The data bar and bottom view will also refresh to the corresponding position.
- The size of the motion parameters corresponds to the depth of the color, and the entire surface space is colored by mapping the motion parameters of the CL point to RGB coloring. In this way, a visual graph of the motion parameters can be obtained.
- The color of the CL points not only compares the before and after CL points, but can also compare the colors of adjacent tool path trajectories.

## 5. Estimation of the Machining Quality of Transition Surface

Due to the fact that the transition surface easily produces machining damage during direction-parallel cutting machining or contour-parallel cutting machining, the actual machining state is simulated directly through the NC device simulation software in order to reduce the trial machining or do away with the need for trial machining. The machining surface quality is predicted by visualization of the machining parameters under the same processing conditions. The damage on the transition surface makes it difficult to establish a mapping relationship with the motion parameters at the corresponding time, and it is also difficult to correspond the local area of the damage to the G code. Based on this, the

simulation software of the NC device is used to collect the motion parameters at the interval of the interpolation cycle under the same machining state. Therefore, the collected motion parameters are colored at the geometric position of the corresponding interpolation point to represent the processing status, such as motion parameters used by the interpolation point. In this way, it is possible to not only record the motion parameters changing with time, but also to compare the rationality of the motion parameters of adjacent tool paths. At the same time, it is possible to directly establish the mapping relationship of the line number of G code where each interpolation point is recorded to the G code.

By running the G code, the binary Biip.dat file is generated, and the mapping relationship between the interpolation point and the motion parameter coloring of the interpolation point is established, and the surface is reconstructed.

The binary Biip.dat file is imported into the self-developed visualization software iSurface to establish the line number and the spatial point mapping of the part, and evaluate the rationality of the surface quality and adjacent trajectory motion parameters.

Therefore, this paper proposes a method for predicting the surface quality of parts with visual motion parameters, and the surface of the part is reconstructed by interpolating points. Based on the generated CL surface, the color gradient processing of the motion parameters is performed to generate a visualization map. The quality of the surface of the part is predicted based on the visualization map.

The interpolation period or shorter position loop and current loop control period are considered as the sampling unit, and geometric information, such as the control position of each axis, is recorded in the NC system. Further, the position of the CL point is reconstructed in the workpiece coordinate system from the control position of each axis, and the surface formed by all the CL points is called the CL surface.

In different stages of processing control, different tool location point positions will be recorded, including the command tool location point position after interpolation, the semi-closed loop feedback tool location point position measured by the motor encoder, and the fully closed loop feedback tool measured by the grating ruler.

The CL point data of different sampling can reconstruct different CL surfaces. As shown in Figure 5, each control cycle can record a command CL point coordinate $p_1$, a feedback CL point coordinate $p_2$, and there is an actual CL point coordinate $p_3$ and a tool cutting CC point coordinate $p_4$. So, the $p_1$ set forms the commanded CL surface, the $p_2$ set forms the feedback CL surface, the $p_3$ set forms the actual CL surface, and the $p_4$ set forms the actual forming surface.

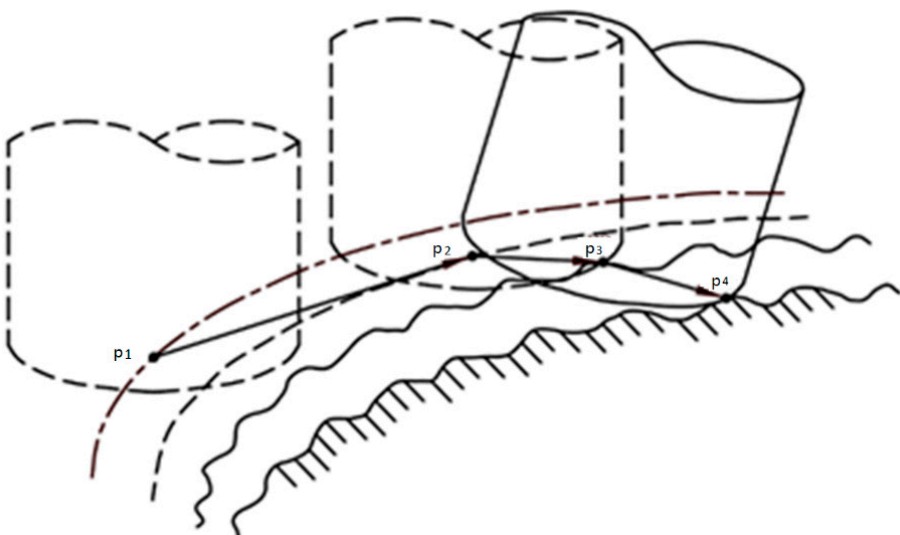

**Figure 5.** The positional relationship of several CL points.

Firstly, the shape of the CL points in space is a surface, and it is necessary to study the discontinuous characteristics of the motion parameters, such as speed on the surface, so as to evaluate the performance of the motion parameters on the surface. Through its expression on the cutter surface, the visualization map of the motion parameters can be obtained. By graying or colorizing the motion parameters, each movement's parameters correspond to a color value, with different values having different colors. The gradient algorithm is used between the maximum and minimum values, so that the motion parameters of the visualization map can be obtained, and its essence is a tool surface map with color.

The speed of the acquisition CL point $P_i$ ($i = 1,..,n$) is $V_i$ ($i = 1, \dots ,n$): the visualization map can be expressed as:

$$V_i \rightarrow RGB(x, y, z), (x \in (0, 255), y \in (0, 255), z \in (0, 255)) \tag{7}$$

By mapping each CL points to RGB coloring, the size of the motion parameters corresponds to the depth of the color, and the whole surface space is colored. The color of the CL point can not only compare the front and after CL points, but also compare the color of adjacent CL points, so as to investigate the rationality of the motion parameters of adjacent CL points. The specific algorithm is as follows in Figure 6:

(1) The machining G code is loaded into the NC device or machine tool;
(2) Motion parameters are collected;
(3) A visualization map of the motion parameters is generated. The size of the motion parameters corresponds to the depth of the color, and the entire space surface is colored by mapping the motion parameters of CL point to RGB coloring. In this way, a visual graph of the motion parameters can be obtained;
(4) Through the visualization map, the quality of the machined surface is predicted. In the surface of variable curvature or transition surface, it is often the inconsistency of adjacent motion parameters that leads to pits and bulges, which is called surface damage defects, and these are much larger than the surface roughness in scale. These damage defects are more common in the transition surface than in the ordinary surface. If not satisfied, go to Step (5), otherwise go to Step (6);
(5) Based on the corresponding relationship between motion parameters and code number, the local feed speed of the machining code is optimized to obtain new machining parameters. Return to Step (1);
(6) The optimized machining parameters and machining codes are output and loaded onto the machine tool for actual machining.

The specific algorithm flow is as follows:

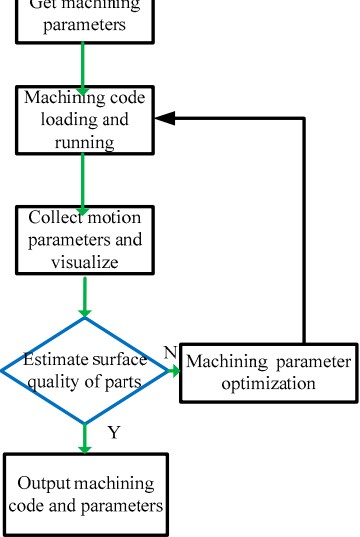

**Figure 6.** Surface quality prediction flowchart.

## 6. Simulation and Experimental Verification

In order to verify the optimization effect of the same speed interval and local trajectory of the transition surface, a section of tool path was selected for verification. At the same time, the Cola bottle bottom and the transition surface with variable curvature were selected to visualize the motion parameters, and the visualization map was generated to verify the effect of surface quality prediction.

### 6.1. Constant Speed Interval Partition Verification

The program segment consisted of 29 CL points, as shown in Figure 7 below. In order to calculate the same speed interval, the error threshold was set $E_{thr} = 0.1$, distance ratio $L_{thr} = 5$, curvature ratio threshold $Q_{thr} = 2$. According to the error threshold $E_{thr} = 0.1$, the characteristic CL points of this segment were obtained, as shown in Figure 7b. The red CL points are the characteristic CL points. In order to obtain the starting point and end point of the same speed interval more accurately, the curvature and distance of the characteristic CL points and the before and after original CL points were calculated. Within the threshold, the characteristic CL points were incorporated into the same speed interval. As shown in Figure 7c, 14 CL points were incorporated into the same speed interval, and four same speed intervals were obtained by dividing the same speed interval.

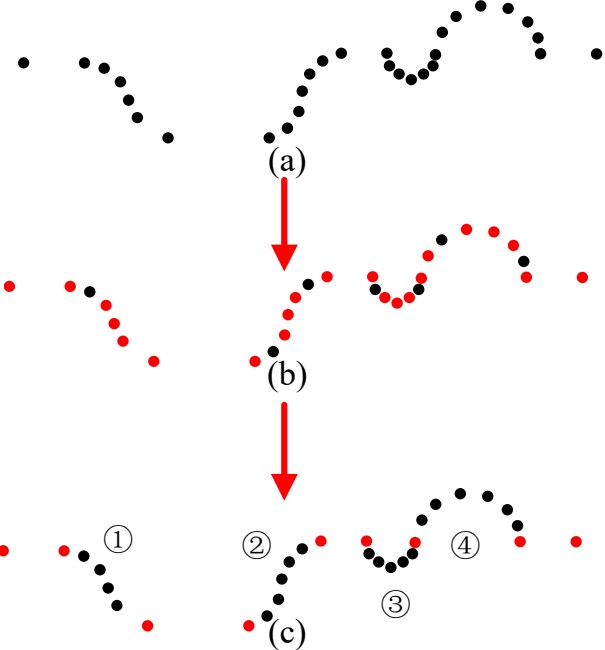

**Figure 7.** Constant interval division (**a**) CL points (**b**) the characteristic CL points (**c**) the same speed interval.

### 6.2. Tool Path Fitting of Akima Spline in the Same Speed Interval

Based on the machining difficulty of the transition surface compared with other surfaces of the same part, the path of the transition surface was locally optimized. In this paper, Akima spline fitting was used, and the CL points of the transition surface ① in Figure 7 were selected to fit into the Akima spline tool path, as shown in Figure 8. By using the method of inputting and fitting at exactly the same time, first of all, the tangent vector of the first and end points were calculated. According to the position information of the first and end points, the Akima spline polynomial coefficients of the tangent vector were calculated by Equation (6), so as to fit the tool path of the first Akima spline, as shown in Figure 8a. The tool paths of the second to the fifth Akima splines were calculated step by step, as shown in Figure 8b,c.

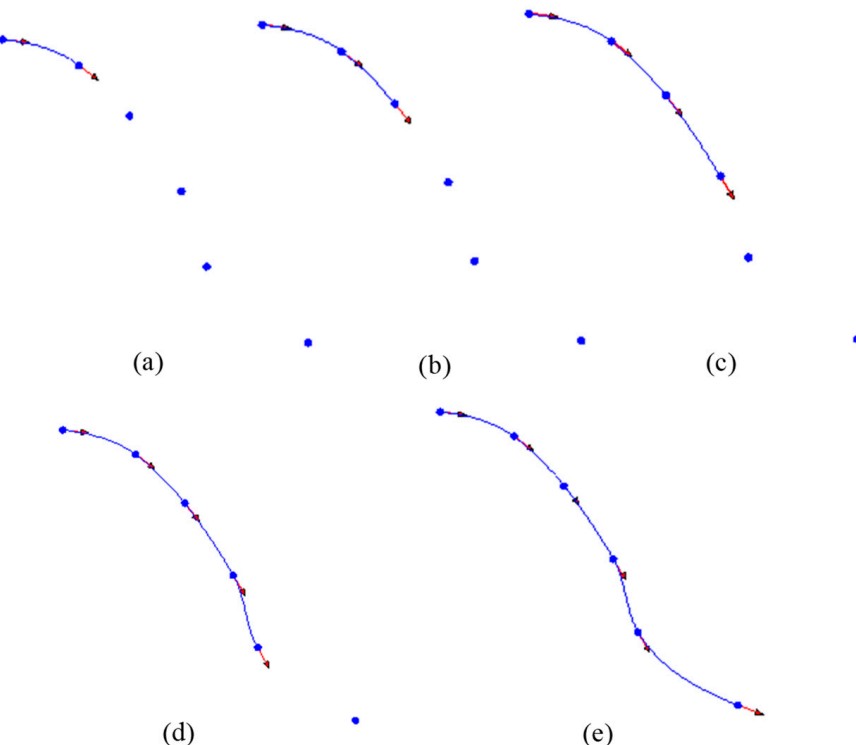

**Figure 8.** Akima spline fitting (**a**) the first Akima spline (**b**) the second Akima spline (**c**) the third Akima spline (**d**) the fourth Akima spline (**e**) the fifth Akima spline.

*6.3. Prediction and Simulation of Machining Quality of Transition Surface*

Taking the bottom part of the Cola bottle as an example, the interpolation file generated by the NC simulation software was imported into the visualization software to display the CL points surface, and the corresponding relationship between the CL point and the tool path was established as shown in Figure 9. The corresponding relationship between the main window, the right-side interpolation data and the motion parameters at the bottom are displayed. The red interpolation points with yellow circle in the curved surface correspond to the trajectory of line 2316 in the G code program, and the corresponding interpolation point was 173,529. It could quickly establish the mapping relationship between the CL point and the line number in the curved surface, which established a good foundation for the subsequent single or local code optimization. In the main window, speed coloring was carried out on the interpolation CL point surface to obtain the speed visualization map, which could visually show the quality of the transition surface of the parts. The bottom corresponded to the speed trajectory, showing the current interpolation point and the feed speed of 2999.8 mm/min.

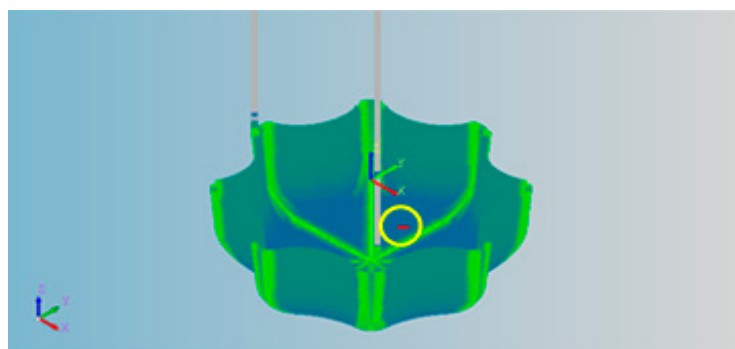

**Figure 9.** Visualization of the bottom of a Coke bottle.

In order to test the quality effect of the predicted transition surface and the improvement effect of the same velocity interval and the spline fitting of the transition surface on the machining of the transition surface, the machining code of the bottom of the Coke bottle was selected. Figure 10 shows that the machining code was not optimized. The interpolation file was generated by the NC machining simulation software, and the visualization map of motion parameters were obtained by inputting the visualization software. It can be seen from the Figure 10 that the motion parameters were not smooth on the transition surface. Figure 11 shows the visualization map of motion parameters obtained after local trajectory optimization and same speed interval planning on the transition surface. In the same area of the red circle, the color was blue, indicating smooth speed. It could be seen from the map comparison that under the same machining conditions, such as the same machine dynamics of machine tool and the same tool, better machining quality could be obtained with a good visualization map.

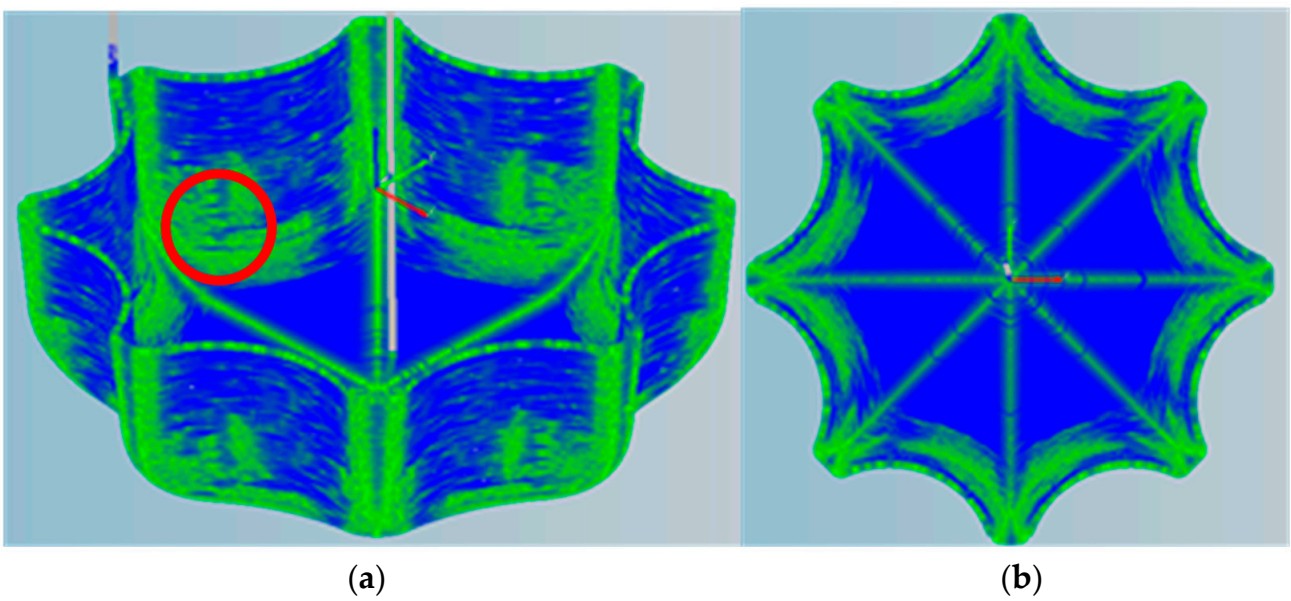

(**a**)          (**b**)

**Figure 10.** Visualization of Coke Bottom before optimization (**a**) before optimization (**b**) after optimization.

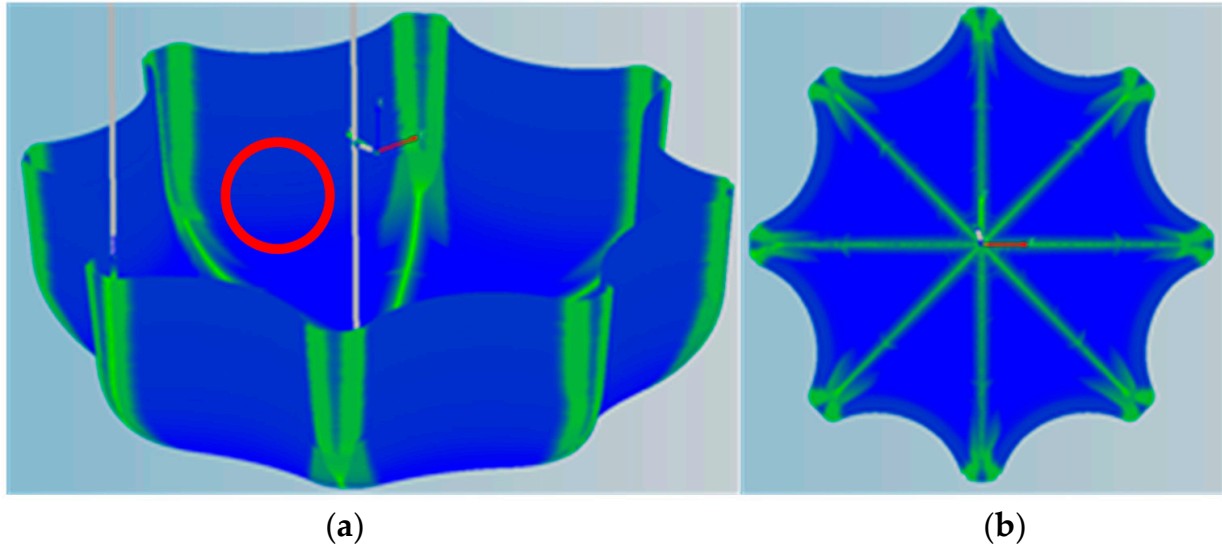

(**a**)          (**b**)

**Figure 11.** Visualization of Coke Bottom after optimization (**a**) before optimization (**b**) after optimization.

Figure 12 shows the speed and acceleration visualization maps before and after the optimization of the transition surface with variable curvature. The visualization maps

significantly improved through the optimization of the transition surface trajectory and speed planning in the same speed interval.

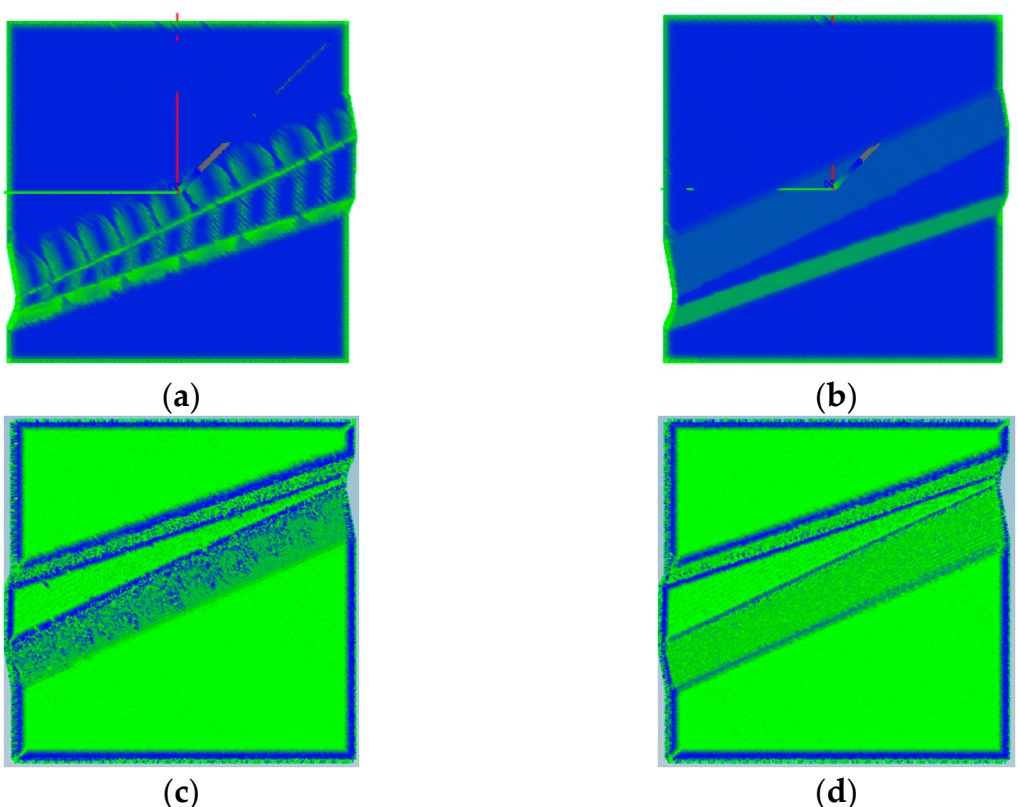

**Figure 12.** Visualization of the transition surface with variable curvature (**a**) speed visualization before optimization (**b**) speed visualization after optimization (**c**) acceleration visualization before optimization (**d**) acceleration visualization after optimization.

In order to verify the consistency between the visualization and the machining quality, the finishing machine tool adopted the Taikan precision T-500S machine tool with the CNC system HNC-818Di, as shown in Figure 13. The cutting tool was D6R3 ball end mill, and the cutting tool was made of cemented carbide. The feed speed was F3000 mm/min, the rotation speed was S12000 r/min, and the cutting depth was 0.08 mm.

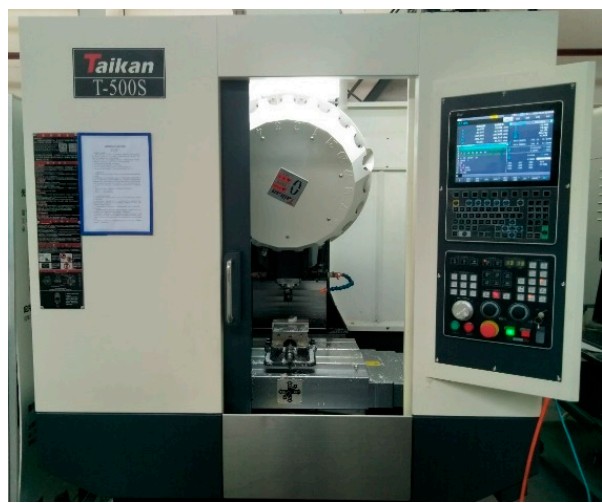

**Figure 13.** Taikan precision T-500S machine tool.

In order to verify the consistency between the visualization map and the machining quality, the parts with variable curvature were processed in the machining center of Baoji Machine Tool, as shown in Figure 14. Under the same machining conditions, the machining time of the code before optimization was 7 min and 58 s, and the machining time after optimization was 7 min and 46 s. Taking into account the detection conditions, three-coordinate travel restrictions and other factors, it was not easy to measure. Through the comparison of roughness standard blocks, the surface roughness of variable curvature before and after optimization was Ra 6.3 and Ra 1.0, respectively, as shown in Table 1. The experimental results showed that, for this kind of transition surface, through local spline fitting, the same speed interval optimization could not only improve the surface quality, but also improved the machining efficiency and shortened the machining time. At the same time, it also showed that the visualization map could better predict the surface quality. When the color of the visual graphics was smooth, the quality was good in the actual machining. Therefore, there was no need for actual machining, and the surface quality could be better estimated through the visualization map.

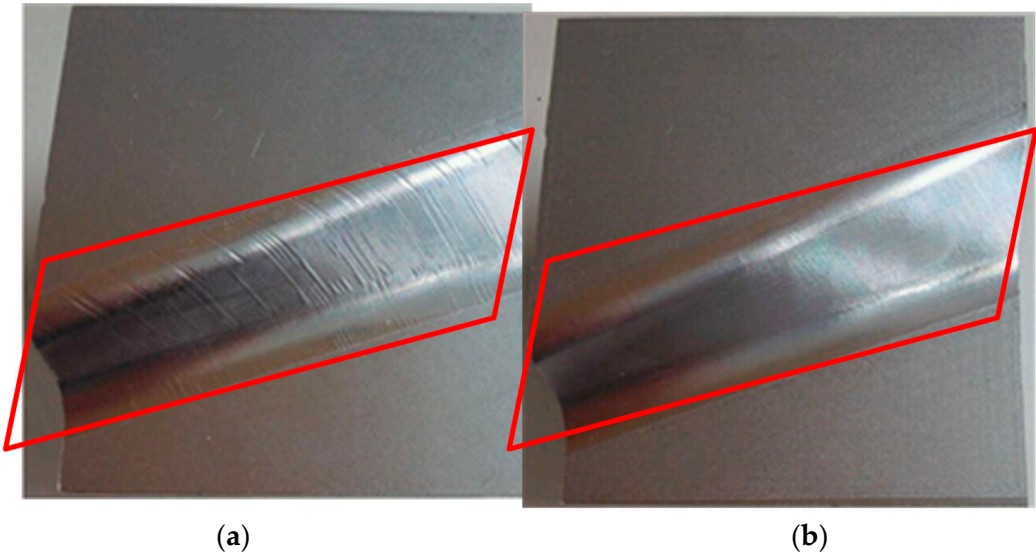

| (a) | (b) |

**Figure 14.** Machining comparison of transition surface with variable curvature (**a**) before optimization (**b**) after optimization.

**Table 1.** Experimental comparison.

| Parts | | Machining Time | Surface Roughness |
|---|---|---|---|
| The variable curvature part | before optimization | 7′58″ | Ra 6.3 |
| | after optimization | 7′46″ | Ra 1.6 |
| The Wuzhishan part | before optimization | 37′26″ | Ra 3.2 |
| | after optimization | 43′49″ | Ra 0.8 |

The following figure shows a Wuzhishan part in Figure 15, which was processed by loop cutting. The quality of the transition surface was provided through the determination of the transition area, the same speed interval and the optimization of the trajectory during the machining. The machining time of the code before optimization was 37 min and 26 s, and the machining time after optimization was 34 min and 49 s. Through comparison of the same roughness standard block, the surface roughness of the Wuzhishan part before and after optimization was Ra 3.2 and Ra 0.8, respectively. For the variable curvature parts and the Wuzhishan parts, no matter whether it was line cutting or circular cutting machining, through the transition surface identification, effective trajectory optimization

and the corresponding same speed interval planning, so as to achieve higher machining speed machining, could improve the quality of surface machining.

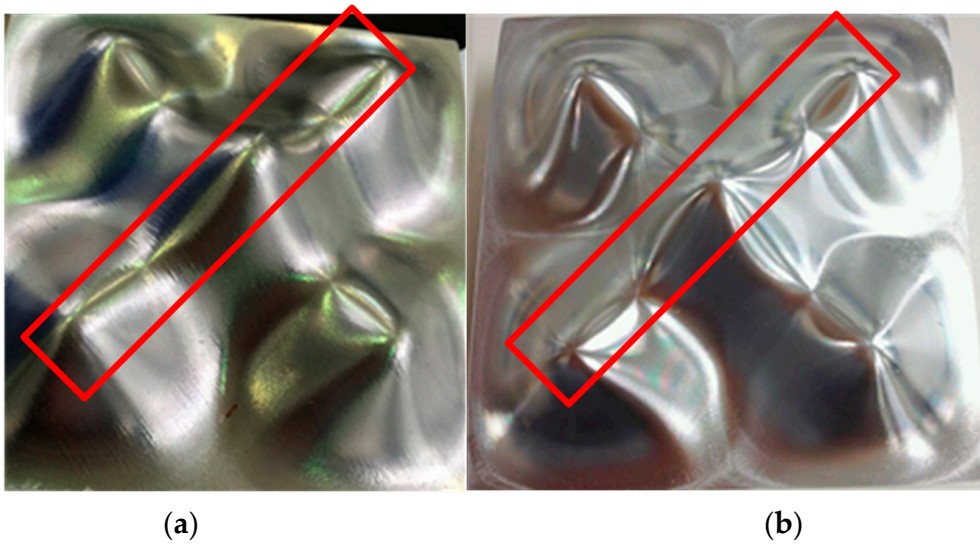

(**a**) (**b**)

**Figure 15.** Machining comparison of Wuzhishan transition surface (**a**) before optimization (**b**) after optimization.

## 7. Conclusions

(1) Considering that the transition surface is more difficult to process than the ordinary surface, adaptive speed planning, local optimization of machining trajectory and visualization of motion parameters during machining were carried out to predict the surface quality of the transition surface. Based on the geometric properties of the tool path of the transition surface, the division method of the same speed interval was established. In the same velocity interval, the Akima spline trajectory with input and fitting was established, which established a foundation for improving the machining quality of the transition surface.

(2) Setting the machining parameters, the motion parameters (speed, acceleration and current) were collected by NC simulation software, and the mapping relationship between the motion parameters and the geometric position of the transition surface was established. The motion parameters visualization software was developed and the visualization map could be formed intuitively through the interpolation file import software, and the machining surface quality could be estimated by the smoothness of color based on the motion parameters of the map. At the same time, the mapping relationship between the interpolation points and the line number of codes could be quickly established, and the corresponding G code could be optimized.

(3) The effectiveness of the surface quality prediction was verified through simulation and experiment. The same speed interval planning and trajectory optimization of the transition surface of the Cola bottle bottom and the variable curvature surface were carried out respectively, and the visualization maps before and after optimization were obtained. By comparison, it was obvious that the surface motion parameters of the optimized visualization map had better smoothness, which showed the effectiveness of same speed interval planning and trajectory fitting of the transition surface. Further, the effectiveness of the predicted quality was verified. Through the actual machining of the variable curvature transition surface, the improvement of the machining surface quality was consistent with the quality of the visualization map, indicating that quality prediction could be carried out with the visualization map.

### 8. Future Work

The interpolated data is directly used surface quality assessment, e.g., roughness and contour error calculations from interpolated data; surface roughness is directly measured with a roughness instrument, as well as verification of the accuracy of evaluating surface quality.

**Author Contributions:** Conceptualization, K.X. and J.W.; methodology, H.Z.; software, K.X.; validation, J.W., H.Z. and D.F.; formal analysis, J.W.; investigation, K.X.; resources, J.W.; data curation, H.Z.; writing—original draft preparation, K.X.; writing—review and editing, J.W.; visualization, K.X.; supervision, D.F.; project administration, K.X.; funding acquisition, D.F. All authors have read and agreed to the published version of the manuscript.

**Funding:** This work was Supported by the Regional Innovation Foundation of China-Hunan Joint (U19A2072), and the Natural Science Foundation of Hunan Province (2021JJ30678).

**Institutional Review Board Statement:** Not applicable.

**Informed Consent Statement:** Not applicable.

**Data Availability Statement:** Not applicable.

**Acknowledgments:** I would like to take this opportunity to thank the reviewers for their comments and the editors for their revisions to the article.

**Conflicts of Interest:** The authors declare no conflict of interest.

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
