# Peer review of "Research on Speed Planning in the Constant Speed Interval and Quality Prediction on the Transition Surface"

_machines, doi:10.3390/machines10070558_

Round 1

Reviewer 1 Report

The article presents an approach to the analysis of transition surfaces in terms of surface quality improvement. The topic is known and often topical in machining analysis, especially in the strategy used for CAM systems. The article is methodological correctly written, the research analysis is clear, it is necessary to make a few clarifications and corrections as follows:
I recommend the authors to make some major changes in abstract:
The abstract is a little bit confuse and missis some information like more results and conclusions, I suggest to authors follow these rules:
- One or two sentences on BACKGROUND
- Two or three sentences on METHODS
- Less than two sentences on RESULTS
- One sentence on CONCLUSION
The introduction section is written coherently and consistently in content. Figure 1 is out of focus, it does not show that there is an improvement in processing between photos. I recommend that it be removed and the explanation of what the work contains keep only in the text.
Chapter 2's title is misleading, maybe it is better to call it Materials and method and include the subsections inside it.
In the drawing, I recommend that you extract them as a, b or c, in the drawing caption, specify only the descriptions - Fig.2. and others - it must be all normalized.
The presented program windows raise doubts (Figs 6,7,8), it can be seen that their content was edited in a graphics program, and very carelessly. I recommend that you remove these materials, it is not necessary to present this type of "achievements" in a scientific work.
Drawing 9 of very poor quality, quite simple, so you can draw it again without much work.
Figure 9 is of very poor quality, it is quite simple, so whatever it is shown in Figure 13, it is illegible, the visible language is also not appropriate on it.
Technical vocabulary in terms of technological processing requires improvement, line 464 - the term "ball knife" - there is no such name, maybe it meant "ball end mill"? I recommend that you check the text more carefully in this respect.
The work presents the improvement of the machining strategy, the visible visual effect in photo 18 proves it. Do the authors not know the basic methods of assessing the quality of the surface after treatment?
Providing a quantitative index and comparing, for example, the basic roughness parameters in two variants would be more appropriate in a scientific work.
The conclusions are adequate and relate to the effects of work. Very generally, however. comparing the processing times for the surface quality parameters - basic Ra and Rz, would improve the scientific appearance of the work.
The practical verification of the research looks like a technical craft, not a scientific analysis, it needs to be improved.

Reviewer 2 Report

Very well written paper. I only have one comment. Only photos of the surface (Fig.18 Fig.19) are presented in the article as a result of the optimization of machining. The photos should be supplemented with some metrological surface measurements, for example the measurement of the surface roughness index. Comparing surface quality from an image alone is not a scientific approach.

Round 2

Reviewer 1 Report

Paper can be published in present form